# PVL-Cartographer: Panoramic Vision-Aided LiDAR Cartographer-Based SLAM for Maverick Mobile Mapping System

**Yujia Zhang** , **Jungwon Kang** and **Gunho Sohn** *

The Department of Earth and Space Science and Engineering, Lassonde School of Engineering, York University, 4700 Keele Street, Toronto, ON M3J 1P3, Canada; zhang89@yorku.ca (Y.Z.); jkang99@yorku.ca (J.K.)
* Correspondence: gsohn@yorku.ca

**Abstract:** The Mobile Mapping System (MMS) plays a crucial role in generating accurate 3D maps for a wide range of applications. However, traditional MMS that utilizes tilted LiDAR (light detection and ranging) faces limitations in capturing comprehensive environmental data. We propose the "PVL-Cartographer" SLAM (Simultaneous Localization And Mapping) approach for MMS to address these limitations. This proposed system incorporates multiple sensors to yield dependable and precise mapping and localization. It consists of two subsystems: early fusion and intermediate fusion. In early fusion, range maps are created from LiDAR points within a panoramic image space, simplifying the integration of visual features. The SLAM system accommodates both visual features with and without augmented ranges. In intermediate fusion, camera and LiDAR nodes are merged using a pose graph, with constraints between nodes derived from IMU (Inertial Measurement Unit) data. Comprehensive testing in challenging outdoor settings demonstrates that the proposed SLAM system can generate trustworthy outcomes even in feature-scarce environments. Ultimately, our suggested PVL-Cartographer system effectively and accurately addresses the MMS localization and mapping challenge.

**Keywords:** SLAM; localization; mapping; mobile mapping system; spherical camera; panoramic image; LiDAR; IMU; sensor fusion; pose graph

## 1. Introduction

Owing to its ability to generate intricate and accurate 3D maps, visual SLAM has garnered significant interest in recent times. With a diverse array of applications spanning mobile robots, navigation, and semantic mapping, the focus has shifted towards developing dependable localization solutions using an assortment of sensors such as cameras, LiDAR, and IMU. Generally, a dense 3D mapping system is composed of three main elements: (1) sensor pose estimation derived from spatial alignments of consecutive frames; (2) 3D scene reconstruction employing the estimated camera pose and combined points; and (3) loop closure identification and pose graph optimization [1].

Advancements in the field of robotics have greatly benefited from SLAM techniques, which are crucial for enabling robots to autonomously navigate and perform tasks. Among these techniques, visual SLAM, which leverages camera images for motion estimation and mapping, has been widely adopted. Originally used for obstacle detection, LiDAR sensors have also been employed for localization via scan registration algorithms that align point clouds. Feature-based localization in SLAM systems can deliver exceptional accuracy when environments have evenly distributed 2D and 3D features obtained from optical or LiDAR sensors. However, this method may be prone to failure in environments with limited visual features and depth variations. On the other hand, inertial sensors such as IMUs do not face these localization limitations and can consistently estimate motion with high frequency and low latency. Nonetheless, consumer-grade IMUs may exhibit significant drift over time.

By combining various sensors, such as optical or LiDAR odometry estimates, this issue can be addressed, resulting in an accurate and dependable SLAM system. Consequently, incorporating multiple sensor modalities is highly desirable for the development of next-generation SLAM systems.

MMSs have emerged as an essential tool for a variety of applications, including urban planning, infrastructure inspection, and autonomous driving. The success of an MMS hinges on its capacity to precisely self-locate and generate a real-time map of its surroundings. SLAM is a prevalent method that empowers MMS to accomplish this objective. By merging data from disparate sensors, such as LiDAR and cameras, SLAM can accurately determine the motion of the MMS and create a map of the nearby environment. In this regard, we present a Maverick MMS outfitted with a tilted multi-beam LiDAR and a panoramic camera, allowing the system to capture a 360-degree view of the environment. Although the tilted LiDAR offers enhanced point density, coverage, and mapping accuracy [2], it presents challenges for SLAM due to its restricted horizontal coverage. However, by combining the data from both sensors, our system can surmount this limitation and achieve precise and reliable SLAM performance.

In our Maverick MMS, the LiDAR is angled towards the ground to enhance mapping performance, while the panoramic camera provides a broad field of view (FoV) for robust SLAM. Conventional camera configurations with limited FoV often result in feature tracking failures in SLAM, while panoramic vision facilitates long-term feature tracking. Prior research [3–6] has showcased the efficacy of panoramic vision for visual odometry and visual SLAM across diverse scenarios.

Panoramic vision, while a promising sensor for SLAM, currently only yields results up to scale, which are inadequate for numerous practical applications. Prior attempts to generate metric results using GPS [5] or ground control points [6] have faced challenges due to their availability and dependability. In this study, we propose an early fusion technique that combines the panoramic camera and LiDAR sensor for SLAM, facilitating metric scale results without requiring external data [5,6]. Our method leverages LiDAR points to produce absolute scale results by assigning a range value acquired from LiDAR points to visual features. We implement our approach on the flexible visual SLAM framework, Open-VSLAM [7], utilizing panoramic vision. Drawing inspiration from earlier work [8,9] on the early fusion of LiDAR points and visual features, our method demonstrates promising results in accurate SLAM with the Maverick MMS.

Following the early fusion of LiDAR points and visual features, we establish a combined Visual-LiDAR-IMU SLAM system by implementing our intermediate fusion through a so-called pose graph formulation [10]. In these pose graphs, nodes symbolize poses, while edges between them represent spatial information, usually constraints obtained from odometry and loop closures in SLAM systems.

Our research makes a substantial contribution to the field of SLAM by presenting a unique Visual-LiDAR-IMU SLAM system designed to fuse multiple sensors. The system is composed of two subsystems: early fusion and middle fusion. In early fusion, range maps are generated from LiDAR points within a panoramic image space, facilitating the direct augmentation of ranges to visual features. This allows the SLAM system to function with both visual features with and without augmented ranges. In middle fusion, a pose graph [10] is employed to merge camera and LiDAR nodes, while IMU data supply constraints between each node, encompassing camera-camera, LiDAR-LiDAR, and camera-LiDAR constraints. Our research presents four main contributions:

- The proposed SLAM system integrates various sensors, such as panoramic cameras, LiDAR sensors, and IMUs, to attain high-precision and sturdy performance.
- The novel early fusion of LiDAR range maps and visual features allows our SLAM system to generate outcomes with absolute scale without the need for external data sources such as GPS or ground control points.
- The middle fusion technique is another key novelty of our research. Employing a pose graph formulation facilitates the smooth combination of data from various sensors,

and it empowers our SLAM system to deliver precise and reliable localization and mapping outcomes.

- We carried out comprehensive tests in demanding outdoor environments to showcase the efficacy and resilience of our proposed system, even in situations with limited features. In summary, our research contributes to advancing more precise and robust SLAM systems for various real-world applications.

In summary, our research significantly contributes to the advancement of SLAM systems by integrating multiple sensors, enabling absolute scale estimation, utilizing pose graph formulation for fusion, and demonstrating robust performance in challenging environments. These contributions address key challenges in SLAM and open new possibilities for more precise and reliable mapping and localization in various application domains.

This paper presents a comprehensive review of existing SLAM systems in Section 2, emphasizing their limitations and drawbacks. Next, in Section 3, we unveil our innovative panoramic vision-aided Cartographer SLAM system. Additionally, we assess the performance of our proposed system using a custom dataset in demanding outdoor environments and share our experimental findings in Section 4. Lastly, in Section 5, we recap our contributions and underscore the possible impacts of our research on future investigations in this domain.

## 2. Related Work

Section 1 describes our PVL-Cartographer SLAM system as an extended Cartographer that integrates a panoramic camera, tilted LiDAR, and IMU sensors. In this section, we will review some of the state-of-the-art odometry and SLAM methods that are related to our work.

### 2.1. Visual SLAM

Feature-based visual SLAM approaches have been devised to identify and track corner-like visual features, including SIFT (Invariant Feature Transform) [11], SURF (Speeded Up Robust Features) [12], and ORB (Oriented FAST and Rotated BRIEF) [13]. ORB-SLAM2 [14] and ORB-SLAM3 [15] have extended these methods for use with monocular, stereo, and RGB-D cameras, as well as visual-inertial modules, incorporating map reuse, loop closing, and relocalization features. However, feature-based techniques encounter difficulties in finding correspondences in environments with simple or repetitive patterns or featureless scenes, resulting in motion estimation or tracking failures. LSD (Large-Scale Direct monocular SLAM) [16] and DSO (Direct Sparse Odometry) [17] are cutting-edge direct visual odometry methods that tackle this issue with precise pose estimation and 3D reconstruction.

Nonetheless, both monocular feature-based SLAM and direct visual SLAM experience scale ambiguity, which can be addressed using depth data. RGB-D SLAM and ToF (Time-of-Flight) SLAM have been developed to supply depth information alongside images. Previous techniques [18,19] utilized RGB images and depth data for estimating incremental motion, treating it as a 3D feature matching problem. Ref. [20] proposes a method that extracts visual features and estimates initial incremental motion with RANSAC-based alignment, then employs the initial motion to initialize the ICP (Iterative Closest Point) estimation. KinectFusion [21] is a groundbreaking RGB-D SLAM system utilized for real-time tracking and mapping, but it may fail in cases of rapid motion or featureless environments. Visual-inertial fusion [1,22] has been effectively employed to overcome such tracking failures.

### 2.2. Panoramic Visual SLAM

The majority of visual SLAM systems rely on the conventional pinhole camera model, which has a limited field of view and can easily encounter tracking failures due to insufficient features. This can be particularly troublesome in situations involving rapid motion,

changing lighting conditions, or texture-less environments. One promising approach to address this issue is to expand the field of view using fisheye or panoramic cameras.

The OpenVSLAM framework [7] is a versatile visual SLAM solution capable of accommodating various camera models, such as pinhole, fisheye, and panoramic cameras. It comprises three primary modules for tracking, mapping, and global optimization, which are inspired by ORB-SLAM. By utilizing the equirectangular camera model for panoramic vision, OpenVSLAM is able to execute SLAM using panoramic cameras.

RPV-SLAM [23] is a range-augmented panoramic visual SLAM solution that builds on the OpenVSLAM framework by generating ranges for visual features using LiDAR points. This enables the system to enhance the accuracy and robustness of feature tracking in challenging environments. These advancements in the OpenVSLAM and RPV-SLAM frameworks represent significant contributions to the field of visual SLAM and have the potential to enable new applications in robotics, augmented reality, and beyond.

### 2.3. LiDAR SLAM

LiDAR-based SLAM systems have gained increasing attention in recent years due to their ability to provide high-resolution 3D data of the environment. One such system is LOAM (Lidar Odometry and Mapping in real-time) [24], which has shown promising results without the need for precise range data. Proposed in 2014, it remains one of the best-performing methods according to the KITTI odometry benchmark dataset [25]. V-LOAM [26], the vision-aided version of LOAM, is its main competitor. LOAM achieves real-time performance by breaking down the odometry problem into high and low-frequency algorithms that work together. The high-frequency algorithm estimates velocity, while the low-frequency algorithm handles point cloud registration and mapping for finer results. This approach allows LOAM to be fast and computationally efficient, ensuring low drift and precise mapping. LOAM uses point-to-plane ICP registration for point cloud registration and extracts features based on roughness, categorizing them as point and edge features. LOAM-livox [27] is an extended version of LOAM designed for LiDARs with small FoV and irregular samplings. Another recent development is SA-LOAM [28], a novel semantic-aided LiDAR SLAM based on LOAM that incorporates semantics in odometry and loop closure detection.

Ref. [29] presents a system that uses transformations computed from ICP to feed a pose graph structure, which is then used on loop closings to build an optimization problem that provides updates for keyframes selected along the trajectory. These updates correct the map of the environment being built and reduce accumulated errors from ICP odometry. However, this system remains susceptible to local minima in which ICP can converge.

SuMa [30] expands on prior research in laser-driven SLAM by incorporating semantic maps for enhancement. SuMa++ [31] builds upon SuMa, facilitating semantic segmentation of point clouds using laser-based technology. By employing this semantic data, pose estimation accuracy can be heightened in complex and uncertain circumstances. More specifically, SuMa++ utilizes semantic coherence between scans and the map to eliminate dynamic objects and offer higher-level constraints during the ICP procedure. This enables the system to integrate both semantic and geometric information solely from three-dimensional laser range scans, resulting in significantly improved pose estimation accuracy. Moreover, in contrast to other SLAM techniques, SuMa++ does not rely on any data derived from visual images.

### 2.4. Sensor-Fusion-Based SLAM

V-LOAM [26] is an innovative extension to LOAM that integrates vision-based components to enhance its performance. Proposed by the same research group, it has been shown to outperform LOAM in certain scenarios according to the KITTI benchmark. V-LOAM is particularly effective in detecting sudden and sharp motions, which can be challenging for traditional Lidar-based odometry systems. The high-frequency module of V-LOAM leverages visual features to estimate the velocity of the vehicle while the Lidar ensures pre-

cision in smaller movements. Additionally, the point set registration and motion estimation refinement are performed in parallel at a lower rate to achieve accurate and efficient results.

Ref. [32] introduced LeGO-LOAM, a lightweight and ground-optimized LiDAR odometry and mapping method based on LOAM. Building upon their previous work, the authors extended LeGO-LOAM to include IMU sensors and visual cameras, resulting in tightly-coupled LiDAR-inertial odometry [33] and LiDAR-visual-inertial odometry [34] via smoothing and mapping. This approach leverages the complementary information provided by different sensors to improve the accuracy and robustness of the system while reducing drift and enhancing mapping capability.

Google's Cartographer [35] is widely acknowledged as a real-time solution for indoor mapping. This 2D SLAM system amalgamates scan-to-submap matching with loop closure detection and graph optimization to create globally consistent maps. A local grid-based SLAM technique is used to generate individual submap trajectories. Concurrently, all scans are matched to nearby submaps using pixel-precise scan matching to establish loop closure constraints. Periodic optimization of the constraint graph containing submap and scan poses yields a globally consistent map. The final map is produced as a GPU-accelerated fusion of completed submaps and the current submap, offering a real-time view for the operator. To expand Cartographer for 3D SLAM, an IMU is needed to gauge gravity and determine the z-direction. Roll and pitch, ascertained from the IMU, are utilized in the scan matcher to minimize the search window in three dimensions. Multiple studies [36,37] have built upon Google Cartographer to enhance its processing speed and precision.

The PVL-Cartographer is an extension of Google's Cartographer SLAM system, incorporating panoramic visual odometry capabilities by integrating panoramic cameras, tilted LiDAR, and IMU sensors to improve overall performance. The proposed sensor-fusion-based SLAM offers several advantages. The integration of data from different sensors helps to compensate for the limitations and uncertainties of individual sensors, leading to more precise, reliable, and robust localization and mapping results. Furthermore, by combining these data sources, the SLAM system can create high-resolution 3D maps with rich geometric and semantic information. Finally, the proposed SLAM can reduce or eliminate the need for external data sources, such as GPS or ground control points. By integrating data from multiple sensors, the system can estimate its position and build a map independently, making it more self-contained and suitable for scenarios where access to external data may be limited or unreliable.

## 3. Methodology

In this section, we introduce the PVL-Cartographer, which leverages a sensor fusion approach to enhance the accuracy and robustness of the SLAM system. The PVL-Cartographer system integrates data from panoramic cameras, tilted LiDAR, and IMU sensors to create a comprehensive and reliable representation of the environment. This multi-sensor approach allows the system to overcome the limitations of individual sensors and provide better performance in challenging scenarios.

### 3.1. Mobile Mapping System

MMS has become increasingly popular in recent years due to its ability to provide geospatial data while the platform is in motion. MMSs typically consist of high-resolution cameras and LiDAR as primary sensors for data acquisition, along with other sensor suites such as the global navigation satellite system (GNSS) and IMU for positioning and geo-referencing. Although MMSs are primarily used for mapping purposes, they are not typically used for odometry, and post-processing is often required to obtain accurate geo-referencing information.

Recent advancements in MMS technology, such as machine learning, artificial intelligence, object extraction, and autonomous vehicles, have driven the development of increasingly sophisticated MMS systems. However, such systems are often limited to outdoor environments due to the inability to collect accurate GNSS data indoors. To ad-

dress this issue, the proposed PVL-Cartographer SLAM system for MMS allows for both localization and mapping in GPS-denied environments. Ref. [38] provides a comprehensive overview of recent MMS technologies, including the types of sensors and platforms utilized in MMS and their capabilities and limitations.

There are numerous industrial MMS products available in the market. Each product offers its own features, specifications, and applications, catering to diverse needs in fields such as geospatial mapping, transportation planning, and asset management. Here are some examples of industrial MMS products:

- Trimble MX series: such as Trimble MX7 and Trimble MX9.
- Leica Pegasus: including models such as Leica Pegasus Two, Leica Pegasus: Backpack, and the latest addition, Leica Pegasus TRK.
- RIEGL VMX series: includes models such as the RIEGL VMX-1HA and VMX-RAIL.
- MobileMapper series by Spectra Precision: including models such as the MobileMapper 300 and MobileMapper 50.
- Velodyne Alpha Prime.

### 3.2. Maverick MMS and Notation

Our study presents a Cartographer-based Panoramic-Visual-LiDAR-IMU SLAM system, utilizing the Maverick MMS as depicted in Figure 1. The MMS is outfitted with a Ladybug5 panoramic camera, a Velodyne HDL-32E LiDAR, and high-precision GPS/IMU. Notably, in our experiments, the GPS data are utilized solely as a ground truth to evaluate the performance of the proposed PVL-Cartographer system. The camera's optical axis is aligned parallel to the ground when mounted on a car, while the LiDAR's spinning axis is tilted at an angle of 45° relative to the ground. During data collection, images are captured at an average rate of 7.5 frames per second, while the LiDAR scans are acquired at a spinning rate of 15 revolutions per second.

In this study, the GPS/IMU coordinate system is designated as the body frame *b*, while *l* and *c* represent the lidar and camera coordinate systems. To ensure accurate sensor fusion, all sensors in the Maverick MMS are calibrated using external parameters, with the X-Y-Z axis pointing forward, right, and downward, as illustrated in Figure 2. We have to perform calibration for all sensors fixed in Maverick MMS before using it.

- IMU calibration: Initially, the IMU was calibrated. The Maverick MMS incorporates the NovAtel SPAN on OEM6, which combines GNSS technology with advanced inertial sensors for positioning and navigation. In the GNSS-IMU system, calibration begins by utilizing initial values from "SETIMUTOANTOFFSET" (or NVM). Subsequently, the "LEVERARMCALIBRATE" command is employed to control the IMU-to-antenna lever arm calibration. The calibration process continues for 600 s or until the standard deviation is below 0.05m, at which point the estimated lever arm converges to an acceptable level.
- LiDAR calibration: To accurately determine the position of the LiDAR relative to the body frame, a selection of control points on the wall and ground was made. Using the mission data collected by Maverick, a self-calibration method was employed, which involved time corrections, LiDAR channel corrections, boresight corrections, and position corrections. The self-calibration process was repeated until the boresight values were corrected to be below 0.008 degrees.
- Panoramic camera calibration: The six cameras are calibrated using Zhang's method [39], which involves capturing checkerboard patterns from different orientations using all cameras. The calibration procedure includes a closed-form solution, followed by a non-linear refinement based on the maximum likelihood criterion. Afterward, LynxView is used to boresight the camera with respect to the body frame. The method involves selecting LiDAR Surveyed Lines and Camera Lines, and formulating the constraint as a non-linear optimization problem to determine the optimum rigid transformation.

**Maverick Mobile Mapping System**

**Sensor Data**

LiDAR Points from a (Tilted) LiDAR of Maverick

Registration between Panoramic Image and Depth from LiDAR

**SLAM Results**

Trajectory

Point Cloud

**Figure 1.** The Maverick MMS with a tilted LiDAR and a panoramic camera.

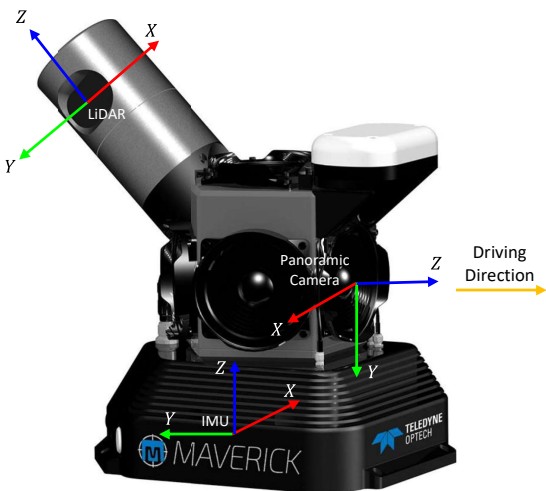

**Figure 2.** The coordinate system for Maverick MMS with a tilted LiDAR, a panoramic camera, and IMU.

A $3 \times 4$ matrix $p = [R,t]$ represents the sensor's pose, where $R$ is a $3 \times 3$ rotation matrix, and $t$ is a translation vector. The variable $k$ denotes a specific time point, where $p_k$ represents the transformation of the local coordinate system at time $k$ relative to its origin. The motion $m_k$ denotes the relative pose between time $k - 1$ and $k$. A 6-DOF pose representation $T_{ab}$ denotes the pose of frame $b$ with respect to frame $a$, so $T_{ab}$ can be used to represent the relative transformation between frame $a$ and frame $b$. The extrinsic parameters of all sensors are listed in Table 1, where the body frame is the IMU frame, $T_{bl}$ denotes the relative transformation between body frame and LiDAR frame and $T_{bc}$ represents the relative transformation between body frame and camera frame.

**Table 1.** Calibration parameters of Maverick MMS.

|  | Rotation Angles [Degrees] | Position [Meters] |
|---|---|---|
| $T_{bl}$ | [179.579305611, −44.646008315, 0.600971839] | [0.111393, 0.010340, −0.181328] |
| $T_{bc}$ | [0, 0, −179.9] | [−0.031239, 0, −0.1115382] |

*3.3. Google Cartographer*

This research presents an enhanced version of the Cartographer SLAM technique, a real-time mapping and loop closure system designed for backpack or vehicle mapping platforms with a 5 cm resolution. The Cartographer method employs a grid-based map representation that permits variable resolution and sensor options. It comprises four modules, as shown in Figure 3. The first module entails data input, which necessitates LiDAR points and IMU data, with odometry pose data and fixed frame pose data as optional extras. The second module contains fundamental processing, which incorporates a voxel filter for LiDAR scan processing. The third module, LiDAR odometry and mapping, employs a Ceres scan matcher for matching features and a submap to estimate the vehicle's pose and orientation. The final global adjustment module optimizes the pose by applying a larger scan matcher to a global map produced by combining all submaps. Cartographer is advantageous because it is a LiDAR-centric SLAM system that can incorporate sensor fusion such as IMU and odometry data.

3.3.1. Local Map Construction

Cartographer generates a map consisting of local submaps and a global map comprised of accumulated submaps. Whenever a loop closure is identified, the global map will be rectified or corrected. Scans from the LiDAR sensor are aligned to a submap coordinate frame iteratively during the construction of a submap. In the submap, the 2D pose transformation of the scan frame $\varepsilon$ is represented as $T_\varepsilon$, as defined by Equation (1):

$$T_\varepsilon p = \underbrace{\begin{pmatrix} cos\varepsilon_\theta & -sin\varepsilon_\theta \\ sin\varepsilon_\theta & cos\varepsilon_\theta \end{pmatrix}}_{rotation} p + \underbrace{\begin{pmatrix} \varepsilon_x \\ \varepsilon_y \end{pmatrix}}_{translation} \qquad (1)$$

Submaps are generated using probability grids. Defined is the pixel that will comprise of all points that are closest to the grid point. A probability hit or miss is ascribed to a set of grid points based on their inclusion in one of these sets. The closest grid point is then added to the hit set for each strike. In the meantime, each miss is added to the grid associated with the pixel, with the exception of grid points that are already included in the hit set.

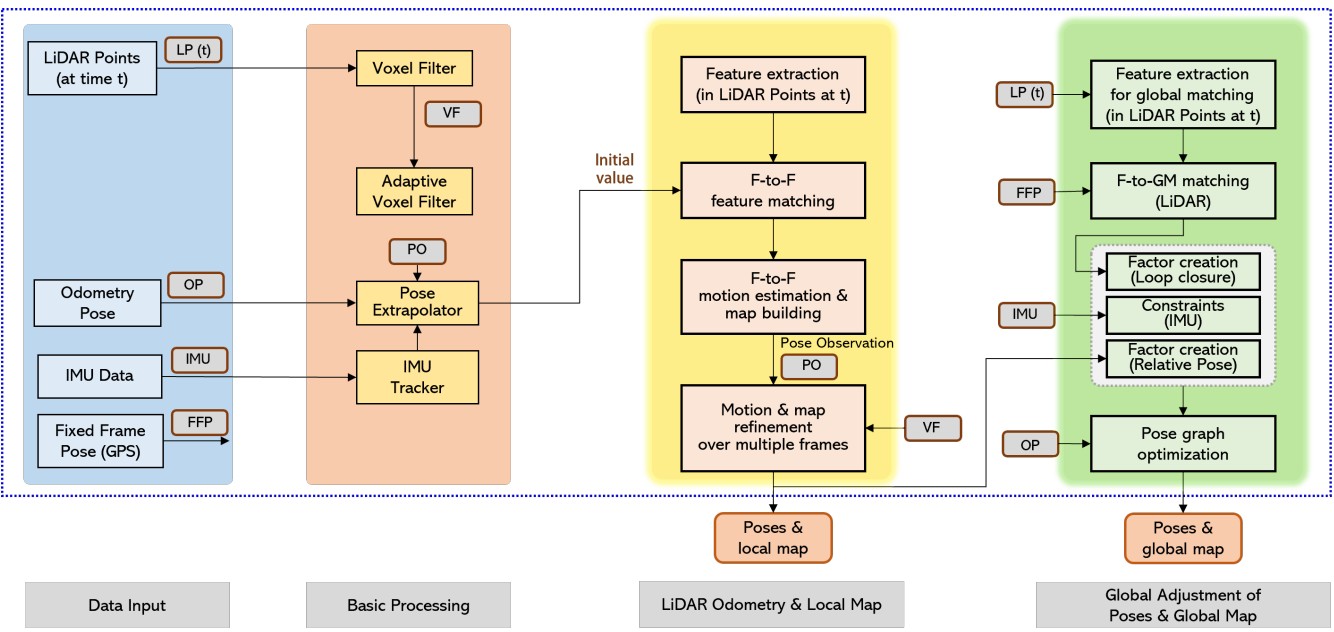

**Figure 3.** The workflow of the Google Cartographer SLAM. The system comprises four modules. The data input module is capable of accepting data from LiDAR, IMU, and odometry (optional), as well as GPS (optional). The second module conducts basic processing of the input data. The third module serves as the frontend of the system, encompassing LiDAR feature extraction, matching, motion estimation, and local map construction. The fourth module functions as the backend of the system, facilitating global map construction based on pose graph optimization.

### 3.3.2. Ceres Scan Matching

In SLAM, the scan matcher is always used to analyze sensor data and estimate position. This approximated pose with translation and rotation is computed by filtering the residual difference between consecutive scans and comparing the current scan states to the submap. The Ceres scan matcher utilized by Cartographer is responsible for determining optimal probability values at the submap's scan points. This is a non-linear least squares problem, as Equation (2) denotes.

$$argmin_\varepsilon \sum_{k=1}^{k} (1 - M_{smooth}(T_\varepsilon h_k))^2 \qquad (2)$$

where $T_\varepsilon$ transforms the scan points $h_k$ from the scan frame to the submap frame based on the scan pose. The function $M_{smooth}$ is a bicubic interpolation smooth filter for the submap's probability values.

This mathematical optimization typically yields greater precision than the grid's resolution. Using the initial estimates from the early fusion, the matching process is made more robust and accurate scan pose estimates are obtained. An IMU is used to approximate the rotational component $\theta$ of the pose between scan matches in the naive Cartographer system. This rotational component is derived in our PVL-Cartographer system from both the IMU and the initial estimates of early fusion.

### 3.4. RPV-SLAM with Early Fusion

In this section, we will present the basis for the range-augmented panoramic visual SLAM (RPV-SLAM) system [23]. The system is composed of four modules: the feature and range module, the tracking module, the mapping module, and the loop closing module. In the feature and range module, visual features are extracted from a panoramic image and their ranges are determined from LiDAR points. The tracking, mapping, and loop-closing modules then assign visual features with and without augmented ranges. At the end

of the pipeline, metrically-scaled results are produced. It is important to note that only results without loop closing are used in the next PVL-Cartographer middle fusion since our PVL-Cartographer SLAM already has its own loop closing module.

### 3.4.1. Feature and Range Module

To begin, we extract ORB features from a panoramic image, which we will call $I$. We then create a range-map, $R$, that is the same size as $I$. Next, we use known calibration parameters to project LiDAR points from the LiDAR frame onto $R$ in the camera frame. These projected points, which we will call $P_l = p_i$, correspond to a point $i$ with a range of $r_i$ at image coordinates $(u_i, v_i)$ in $R$. Next, we need to calculate ranges for the range-map $R$ across the panoramic image area using range interpolation. The LiDAR has a tilted configuration, so the projected points $P_l$ are limited to a rainbow-shaped region (as shown in Figure 1). To obtain a dense range-map $R$ with dense ranges across the panoramic image area, we use Delaunay-triangulation-based interpolation with $P_l$. The interpolated range may become inaccurate if a location $(u, v)$ is far from $P_l$. To address this issue, a binary mask called $M$ is created with the same size as the range-map $R$. The mask keeps the interpolated ranges inside it and discards those outside. The augmentation of ranges to ORB visual features can be performed easily by locating the range in the final generated range-map $R_f$ with respect to the same location as a visual feature. Two types of visual features are extracted as a consequence of the augmentation: (i) visual features augmented with ranges and (ii) visual features without augmented ranges.

### 3.4.2. Tracking Module

The tracking module for localization estimates the camera pose for each frame by finding feature matches to the local map. It uses visual features with and without augmented ranges to achieve this. A motion-only bundle adjustment is performed simultaneously to minimize the reprojection error. The visual features with augmented ranges can be used to create scaled map points directly. However, the scaled map points of visual features without augmented ranges are created using triangulation between two frames under an estimated scaled motion. Once appropriate scaled map points are generated, metrically-scaled SLAM results will be produced.

### 3.5. PVL-Cartographer SLAM with Pose-Graph-Based Middle Fusion

The PVL-Cartographer SLAM has two parts: frontend and backend. Figure 4 shows the workflow of the proposed system. The frontend includes feature extraction, matching, pose estimation, and data association for the local map. Compared to the original Google Cartographer (Figure 3), we added early fusion, combining LiDAR points and panoramic images into the visual feature tracking module. The motion estimation is then used as the initial value in the Google Cartographer LiDAR odometry module. Finally, the camera and LiDAR nodes are inserted into the global map and optimized together in the backend.

First, the pose is generated by the early fusion of the visual feature and range module. The generated pose is then used as the initial value for the modified Cartographer system. Note that any visual odometry algorithm, including RPV-SLAM-based visual odometry, can be applied to the proposed PVL-Cartographer. In this endeavour, we utilized RPV as an early fusion method, which is interchangeable with other visual odometry. This initial value is accepted by the Ceres scan matcher, which then processes the LiDAR points to generate more robust and accurate pose estimates. Next, nodes are added to the global map based on the estimated camera frame poses from early fusion and the estimated LiDAR scan poses from the Ceres scan matcher. Figure 5 depicts the global map structure of the proposed PVL-Cartographer SLAM. The global map has two categories of nodes: camera nodes and LiDAR nodes. All nodes are listed according to the frame or scan's timestamp. Given certain constraints, the nodes in the world coordinate system are optimized. Since the IMU has the highest frequency, it provides measurements of angular velocity and acceleration that can be used to link nodes together. IMU measurements are used to calculate the

constraints between camera frames, LiDAR scans, and consecutive camera-LiDAR nodes. The global map will then be optimized, as detailed in the subsequent section.

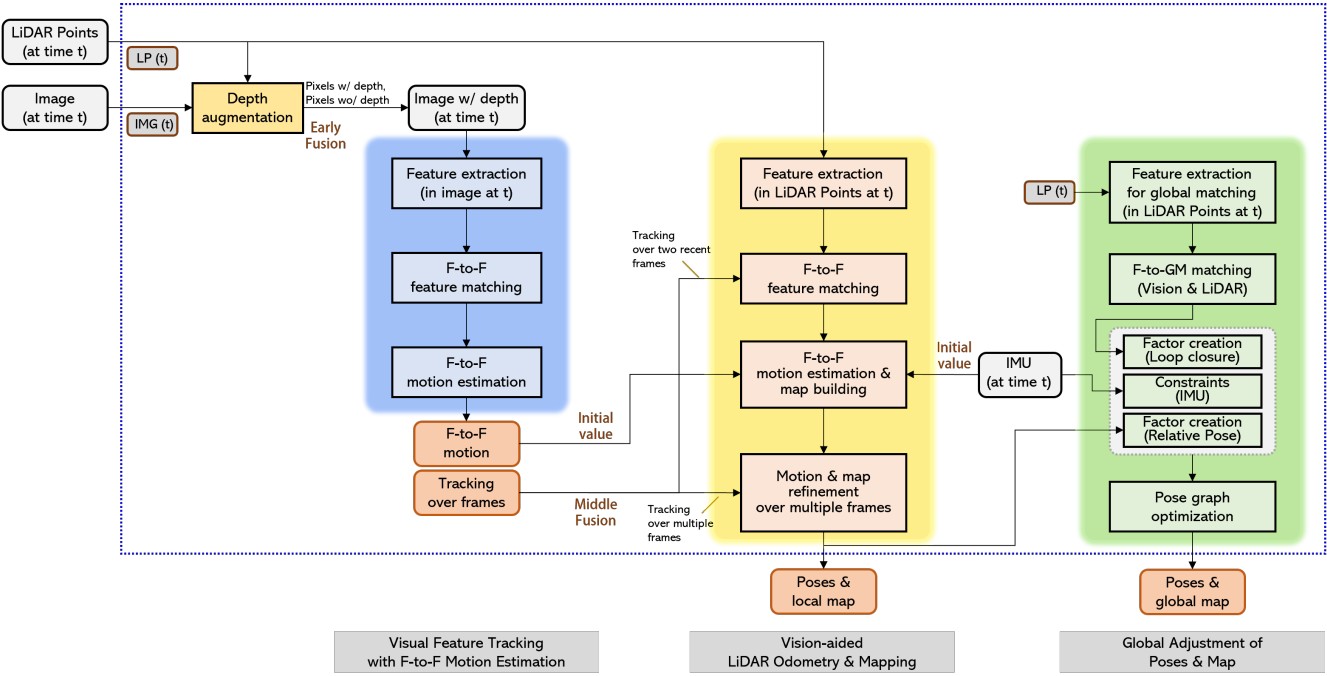

**Figure 4.** The workflow of the PVL-Cartographer SLAM with middle fusion. Compared to the original Google Cartographer, we have added a visual odometry module with early fusion (RPV). The LiDAR odometry based on Google Cartographer incorporates the odometry result from RPV as an initial value. In the backend, we have included camera nodes and LiDAR nodes in the global map. All nodes are optimized with specified constraints. The global map structure is depicted separately in Figure 5.

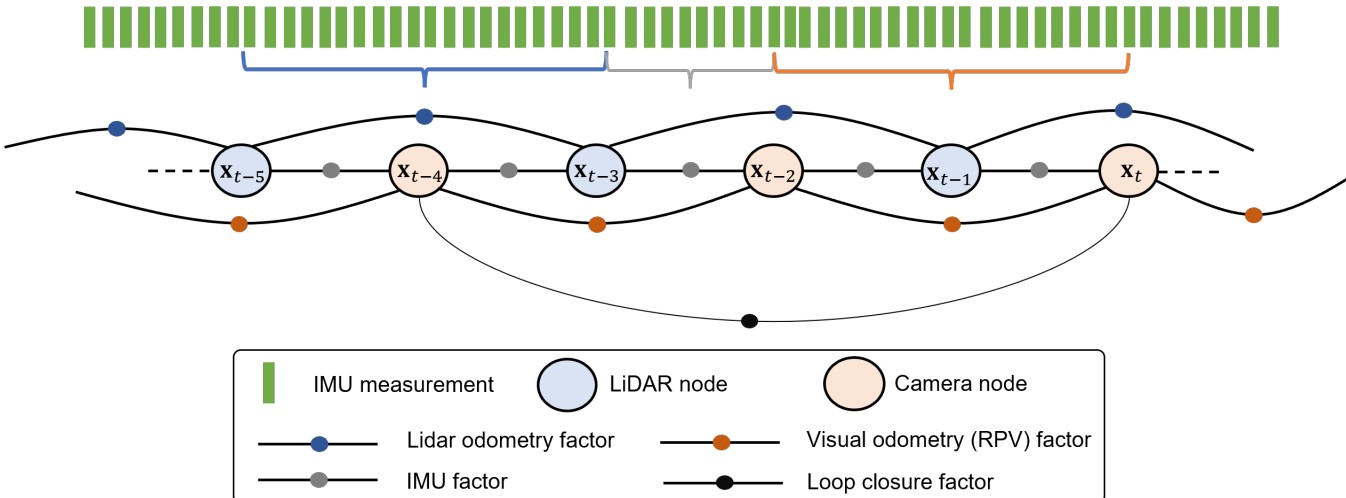

**Figure 5.** The global map structure of the proposed PVL-Cartographer SLAM. The system receives input from a spherical camera, 3D LiDAR, and IMU.

### 3.6. Global Map Optimization and Loop Closure for PVL-Cartographer

In the process of approximating the position in local SLAM, errors are introduced due to the presence of grid map resolution, sensor noises, and the fusion of various map features. Although the error between frames or scans is small, it accumulates and results in a larger drift error after traveling a great distance. Using the sparse pose adjustment

(SPA) optimization procedure, Google's original Cartographer attempts to reduce this error. In this method of optimization, a similar scan matcher in local SLAM is used for pose correction but in a larger global map range. In the interim, loop closure detection is incorporated into the process of optimization. The optimization problem is specified as a non-linear squares problem. Every few seconds, the Ceres is used to compute the solution to (3).

$$argmin_{\Xi^m \Xi^n} \frac{1}{2} \sum_{ij} \rho(E^2(\varepsilon_i^m, \varepsilon_j^n, \sum_{ij}, \varepsilon_{ij})) \tag{3}$$

where the submap poses $\Xi^m = \{\varepsilon_i^m\}_{i=1,...,m}$ and the camera or scan poses $\Xi^n = \{\varepsilon_j^n\}_{j=1,...,n}$ in the world are optimized with given constraints. These constraints take the form of relative poses $\varepsilon_{ij}$ and associated covariance matrices $\sum_{ij}$.

The transformation between two nodes $p_i$ and $p_j$ can be computed by Equation (4).

$$T(p_i, p_j) = \begin{pmatrix} R_{\varepsilon_i^m}^{-1}(t_{\varepsilon_i^m} - t_{\varepsilon_j^n}) \\ \varepsilon_{i;\theta}^m - \varepsilon_{j;\theta}^n \end{pmatrix} \tag{4}$$

Then the residual *E* for such a constraint is computed by Equations (5) and (6).

$$E^2(\varepsilon_i^m, \varepsilon_j^n; \sum_{ij}, \varepsilon_{ij}) = e(\varepsilon_i^m, \varepsilon_j^n; \sum_{ij}, \varepsilon_{ij})^2 \sum_{ij} e(\varepsilon_i^m, \varepsilon_j^n; \sum_{ij}, \varepsilon_{ij})^2 \tag{5}$$

$$e(\varepsilon_i^m, \varepsilon_j^n; \sum_{ij}, \varepsilon_{ij}) = \varepsilon_{ij} - \begin{pmatrix} R_{\varepsilon_i^m}^{-1}(t_{\varepsilon_i^m} - t_{\varepsilon_j^n}) \\ \varepsilon_{i;\theta}^m - \varepsilon_{j;\theta}^n \end{pmatrix} \tag{6}$$

The residuals of camera-camera, LiDAR-LiDAR, and camera-LiDAR are computed separately and then combined as Equation (7).

$$argmin_{\Xi^m \Xi^n} \frac{1}{2} \sum_{ij} \rho(E_{c-c}^2 + E_{l-l}^2 + E_{c-l}^2) \tag{7}$$

The approximated positions from early fusion can also be viewed as landmarks on the globe map. However, the camera frame and LiDAR scan are captured at separate times. This asynchronization issue is solvable through interpolation.

The complete weighted landmark cost function can then be represented by Equation (8).

$$f(p_0^l, p_i^c, p_j^c) = f(p_0^l, p_0^c) = (w_t \quad w_r)(T_{cl}^m - T(p_0^l, p_0^c)) \tag{8}$$

The translation and rotation weights $w_t, w_r$ are included in the landmark observation data, and $T_{cl}^m$ is the transformation between the camera and LiDAR, which has a fixed value for the Maverick MMS.

## 4. Experiments

In this section, we evaluate the proposed SLAM system using an MMS in real-world outdoor environments.

### 4.1. Dataset

We conducted experiments utilizing the Maverick sensor, which was attached to a vehicle and tested in various outdoor settings, such as parking lots, roads, campuses, and residential areas. In order to accurately evaluate the effectiveness of our PVL-Cartographer SLAM system, we selected four specific sets of data. The information for these sets can be found in Table 2, which displays the number of camera and LiDAR frames, image size, distance traveled, running time, scene descriptions, and evaluation methods. To ensure precise ground-truth trajectories, we obtained cm-level accurate GPS data that were processed through bundle adjustment with ground control points in the LMS software from Teledyne

Optech. We successfully synchronized data from multiple sensors using high-precision GPS/IMU data, and LiDAR points through offline interpolation of the timestamps and bundle adjustment method. Then we processed the GNSS data to estimate the trajectory of the system. Base stations were used to improve the accuracy of the GNSS measurements by applying real-time kinematic (RTK) error correction techniques. Furthermore, Figure 6 overlays the ground-truth trajectories on satellite images, giving a clear visual representation of the data and the accuracy of our results.

**Table 2.** Details of our four datasets captured by Maverick MMS to evaluate different methods.

|  | Sequence A | Sequence B | Sequence C | Sequence D |
|---|---|---|---|---|
| Sensors | Maverick MMS: Ladybug-5 + Velodyne HDL-32 + IMU | | | |
| Region | Parking lot | Campus area | Residential area | Residential area |
| Camera frames | 717 | 8382 | 10,778 | 4500 |
| Image size | 4096 × 2048 | 8000 × 4000 | 8000 × 4000 | 8000 × 4000 |
| LiDAR frames | 1432 | 17,395 | 22,992 | 9615 |
| Distance travelled | 324 m | 7035 m | 7965 m | 3634 m |
| Running time | 94 s | 19 min | 22 min | 10 min |
| Ground truth | GNSS/IMU | GNSS/IMU | GNSS/IMU | GNSS/IMU |
| Loop | One small loop | One large loop + a few small loops | Many medium-size loops | A few loops |
| Dynamic objects | Parking, barrier and person | Car, bus and person | Car, bus and person | Car, bus and person |
| Compared methods | ORB-SLAM2 (camera-only) VINS-Mono-SLAM (camera + IMU) LOAM (LiDAR) Google-Cartographer-SLAM (LiDAR + IMU) RPV-SLAM (Panoramic camera + LiDAR) Our PVL-Cartographer SLAM (Panoramic camera + LiDAR + IMU) | | | |

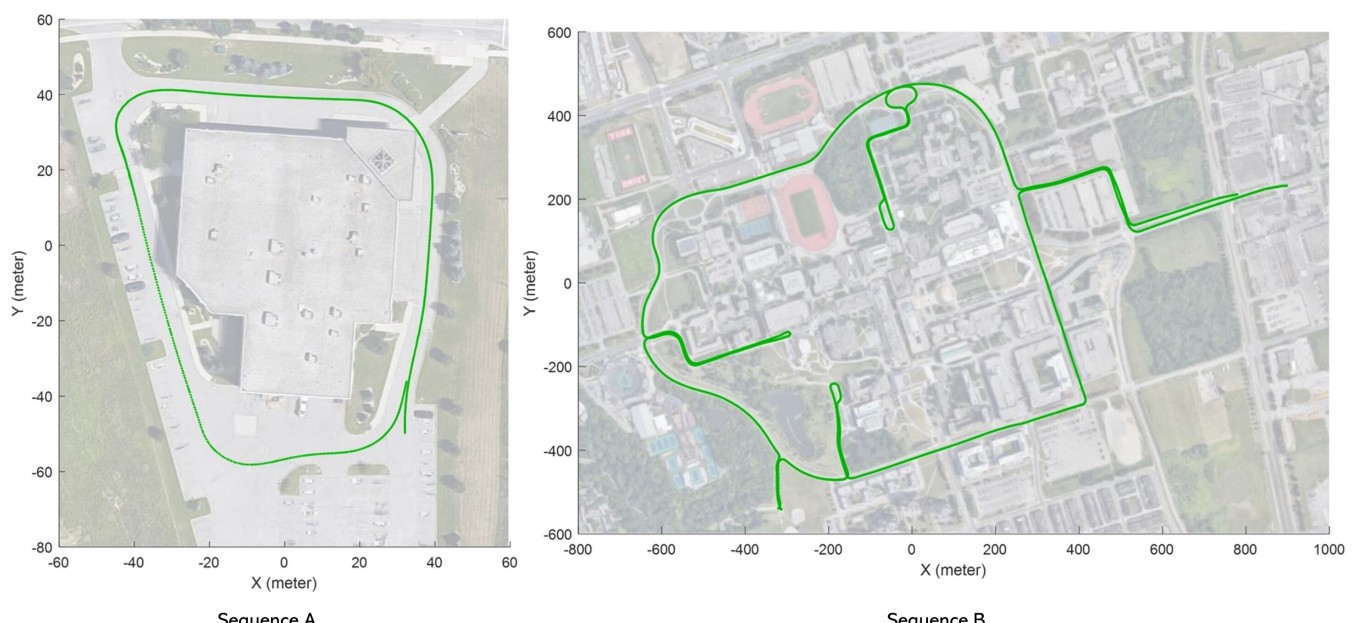

Sequence A

Sequence B

**Figure 6.** *Cont.*

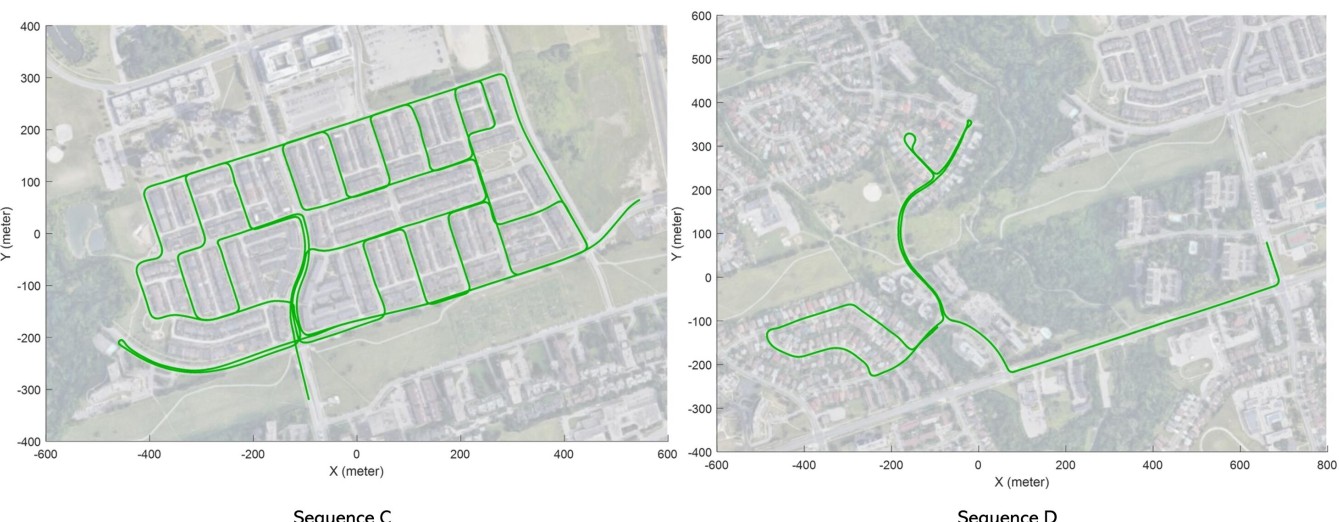

<div align="center">Sequence C         Sequence D</div>

**Figure 6.** Ground–truth trajectories (marked by green dots) overlaid on satellite images for the sequence A, B, C, D.

### 4.2. Results

We evaluated the performance of the proposed PVL-Cartographer SLAM system by comparing it with several state-of-the-art SLAM systems in Table 2. This includes ORB-SLAM2 [14], VINS-Mono [40], LOAM [24], and Google Cartographer [35]. It is worth noting that we tested ORB-SLAM2 with monocular images to highlight the benefits of using panoramic images in visual SLAM. We also tested VINS-Mono, a visual-inertial SLAM system, using the monocular images and IMU data from our Maverick MMS. Additionally, we evaluated LOAM, a well-known LiDAR-based odometry system, and Google Cartographer, a LiDAR-inertial SLAM system that uses IMU data to define the z-direction.

The results indicate that both the RPV-SLAM and proposed PVL-Cartographer systems are capable of producing accurate and robust results on a large scale. Figure 7 displays the trajectories and global maps generated by sensor-fusion-based SLAM systems. Respectively, Figure 8 shows the position errors in the x, y, and z directions for three methods and four sequences. Notably, the initial values of the PVL middle fusion module are derived from the RPV early fusion module, thereby enhancing the PVL-Cartographer system's precision. Incorporating the camera nodes and LiDAR nodes into the pose graph and optimizing them jointly, the proposed PVL-Cartographer system obtains superior performance compared to existing methods.

To evaluate the performance of the proposed PVL-Cartographer SLAM system and the compared SLAM methods, we used the RPG trajectory evaluation toolbox [41] and measured the absolute trajectory error (ATE), relative translation error (RTE), and relative rotation error (RRE) for each method. The ATE is presented in Table 3 as the measured root mean square error (RMSE) using the aligned estimation and the ground truth. The RTE is the average transnational RMSE in percentage over trajectory segments with lengths of 10%, 20%, 30%, 40%, and 50% of the total length. In contrast, the RRE is the average rotation RMSE ($^\circ$/m) over the same trajectory segments. The RTE and RRE outcomes are depicted in Table 4 and Figure 9, respectively.

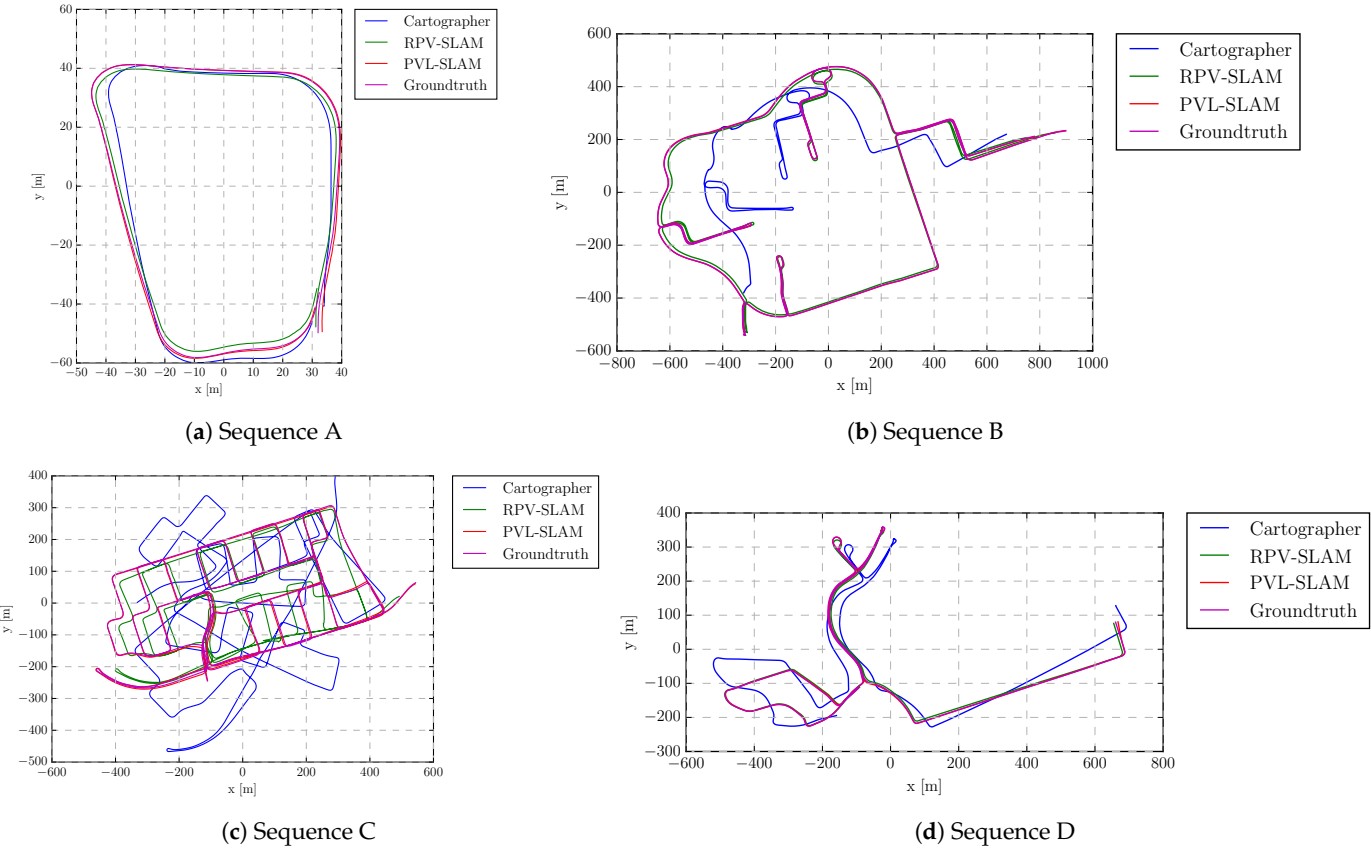

**Figure 7.** Trajectory comparison in each sequence. Note that only a partial trajectory of the Cartographer is shown in (**b**), as the operation of Cartographer was suspended in the middle of the sequence.

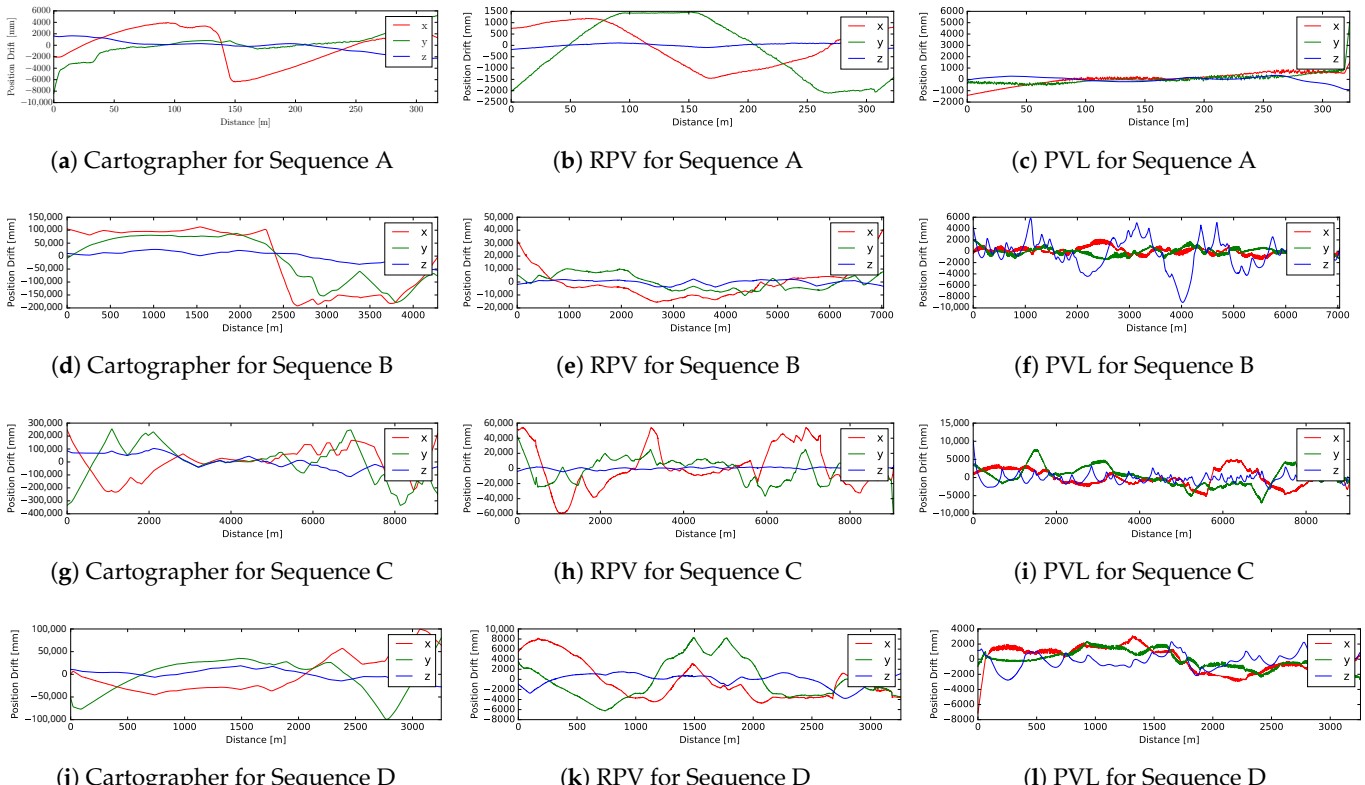

**Figure 8.** Comparison of position errors in x-y-z directions, respectively, for Figure 7.

**Table 3.** Comparison of translation accuracy w.r.t. ATE [m]. The best results are highlighted in bold.

|  | **ORB SLAM2** | **VINS-Mono** | **LOAM** | **Cartographer** | **RPV-SLAM** | **PVL-SLAM** |
|---|---|---|---|---|---|---|
| Sequence A | 5.894 | 3.9974 | Fail | 4.023 | 1.618 | **0.766** |
| Sequence B | 100.870 | 86.897 | Fail | 152.230 | 12.910 | **2.599** |
| Sequence C | 155.908 | 160.765 | Fail | 183.619 | 30.661 | **3.739** |
| Sequence D | 10.665 | 12.875 | Fail | 58.576 | 5.673 | **2.204** |
| Overall | 68.3343 | 66.1336 | Fail | 99.612 | 12.7155 | **2.327** |

**Table 4.** Comparison of translation and rotation accuracy w.r.t. relative error, [%] [deg/m] . The best results are highlighted in bold.

|  | **ORB SLAM2** | **VINS-Mono** | **LOAM** | **Cartographer** | **RPV-SLAM** | **PVL-SLAM** |
|---|---|---|---|---|---|---|
| Sequence A | 7.769<br>0.0677 | 4.685<br>0.0410 | Fail | 6.789<br>0.0507 | 3.934<br>**0.0040** | **3.027**<br>0.0236 |
| Sequence B | 13.770<br>0.0099 | 10.779<br>0.0109 | Fail | 15.047<br>0.0090 | 3.096<br>**0.0009** | **1.273**<br>0.0019 |
| Sequence C | 4.879<br>0.0289 | 3.987<br>0.0301 | Fail | 5.764<br>0.0133 | 3.752<br>0.0057 | **0.853**<br>**0.0018** |
| Sequence D | 2.878<br>0.0148 | 3.085<br>0.0178 | Fail | 4.650<br>0.0137 | **1.347**<br>**0.0017** | 2.555<br>0.0035 |
| Overall | 7.324<br>0.030 | 5.634<br>0.025 | Fail | 9.843<br>0.059 | 2.393<br>**0.002** | **1.069**<br>0.003 |

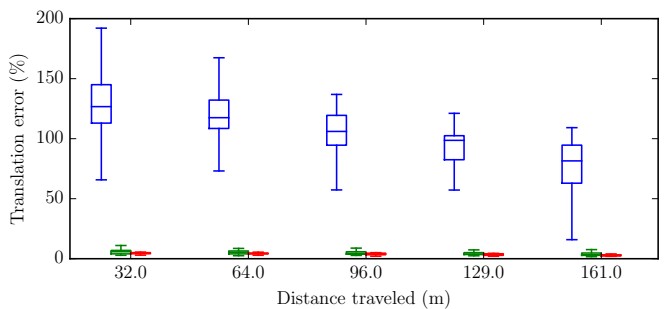

(**a**) Relative translation error for sequence A

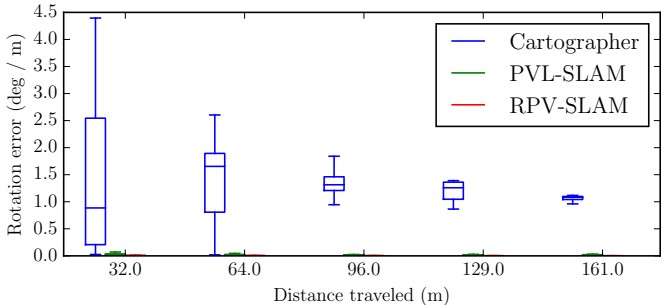

(**b**) Relative rotation error for Sequence A

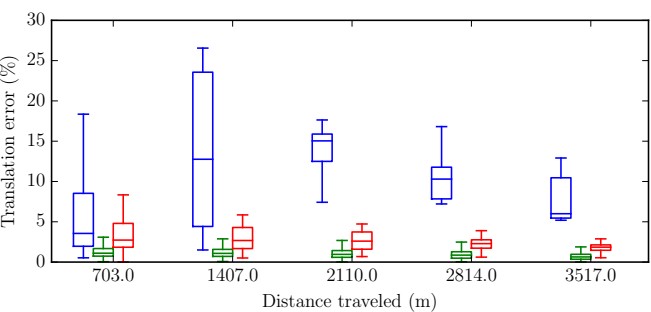

(**c**) Relative translation error for sequence B

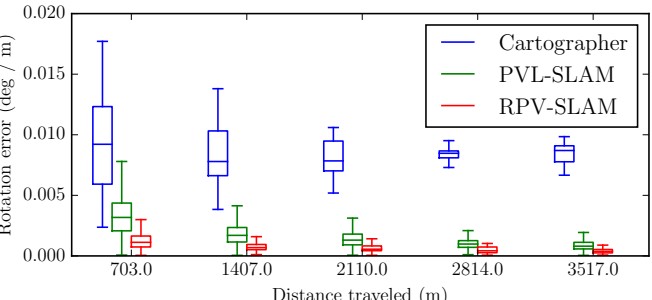

(**d**) Relative rotation error for Sequence B

**Figure 9.** *Cont.*

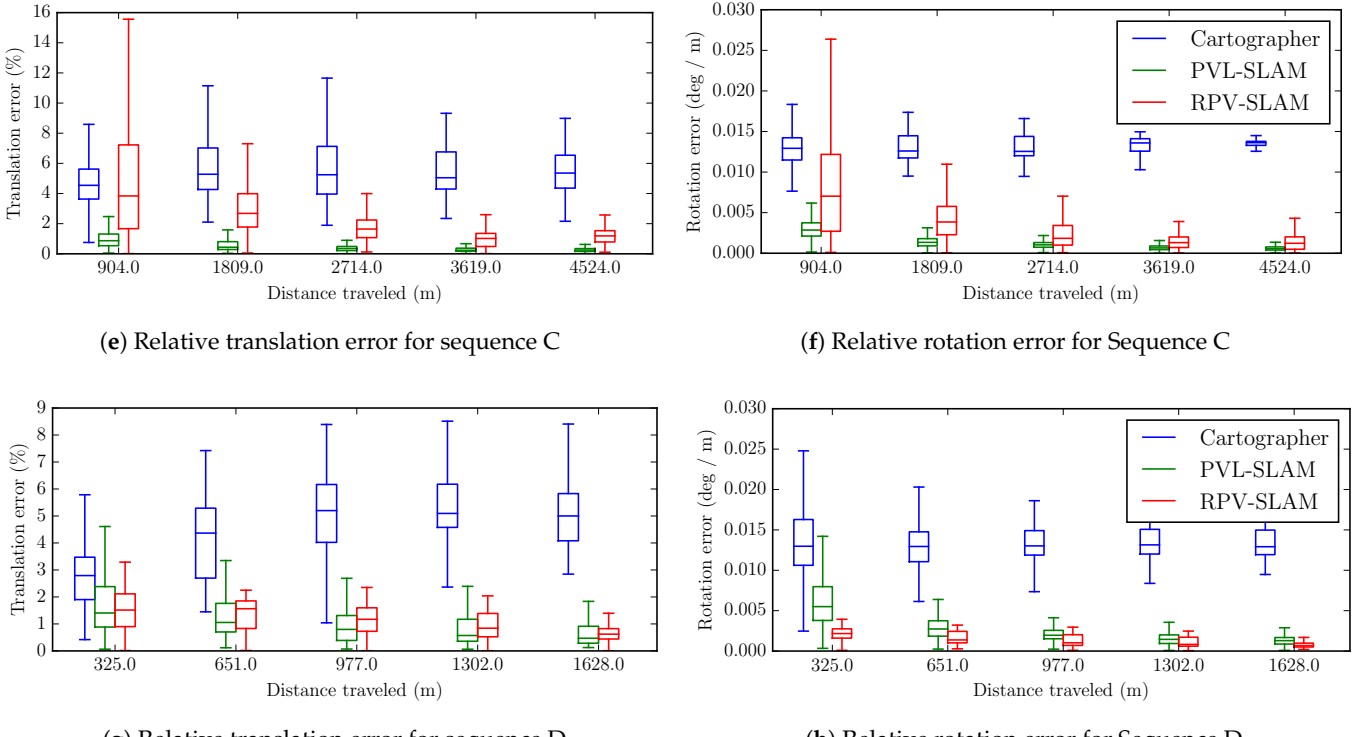

(**e**) Relative translation error for sequence C　　　　　　(**f**) Relative rotation error for Sequence C

(**g**) Relative translation error for sequence D　　　　　　(**h**) Relative rotation error for Sequence D

**Figure 9.** Relative translation and rotation errors for different sub-trajectory lengths shown as a series of boxplots.

### 4.3. Discussion

The proposed PVL-Cartographer SLAM system surpassed Google Cartographer in all four tested sequences, enhancing ATE by 80.96%, 98.29%, 97.96%, and 96.24%, respectively. Moreover, our system outperformed Google Cartographer in terms of RTE by 55.41%, 91.54%, 85.20%, and 45.05%, and in terms of RRE by 53.45%, 78.89%, 86.47%, and 74.45% for sequences A through D. For RRE, it is noteworthy that RPV-SLAM surpassed our PVL-Cartographer. Our findings suggest that camera-centric SLAM systems, such as RPV-SLAM, excel in orientation estimation, while LiDAR-centric systems perform better in translation estimation. Sequence C, with its long distance and intricate loop structures, posed the most significant challenges among the collected data for various scenarios. Nevertheless, our PVL-Cartographer system achieved the best performance in ATE, RTE, and RRE, thanks to the highly effective loop closure module, which dramatically reduced errors through map optimization when multiple loops were detected. Despite the challenges posed by this scenario, our PVL-Cartographer system displayed robustness and superior performance.

The RPV and PVL-Cartographer SLAM systems proved to be more accurate and robust than existing methods, demonstrating the efficacy of sensor-fusion-based SLAM. Figure 9 showed that translation and rotation errors diminished as distance increased, indicating the strong performance of the loop-closure model over extended distances. Additionally, Figure 10 illustrated that the proposed SLAM system could successfully generate precise trajectories even in demanding scenarios with complex loops and long distances. In contrast, ORB-SLAM2, VINS-Mono, and Cartographer could only produce partial trajectories due to insufficient features for matching, while LOAM generated incorrect trajectories, particularly regarding orientation angles, when a tilted LiDAR only scanned points, and no IMU was employed to indicate the z-direction.

Our results revealed that the proposed sensor-fusion-based SLAM systems were more robust than camera-only and LiDAR-only approaches, especially for challenging data collected by an MMS equipped with a panoramic camera and a tilted LiDAR. The out-

comes also demonstrated that the PVL-Cartographer system, integrating panoramic visual odometry, tilted LiDAR, and IMU sensors, delivered accurate and robust results on a large scale.

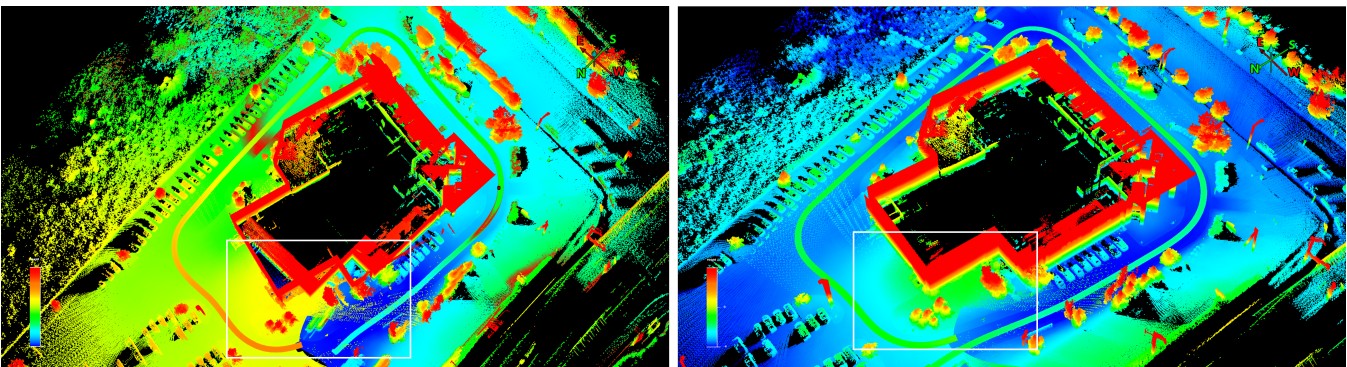

**Figure 10.** For Sequence A, the **left** image shows the misalignment without loop closure, and the **right** image shows the loop closing result. The area enclosed by the white rectangle is where loop closure detection should occur.

## 5. Conclusions

The PVL-Cartographer-SLAM is a sensor-fusion-based SLAM system designed for a Maverick MMS equipped with a panoramic camera and tilted LiDAR. Our approach securely integrates multiple sensors, such as a panoramic camera, LiDAR, and IMU, within a pose graph to enable robust and accurate SLAM. The system includes an early fusion range augmented panoramic visual odometry system, RPV, which produces metrically-scaled trajectories by combining visual features with ranges obtained from LiDAR points. Experiments demonstrate that our proposed system surpasses existing state-of-the-art SLAM systems, including camera-centric, camera-inertial, LiDAR-centric, and LiDAR-inertial SLAM systems, even when only a limited number of visual features are augmented with ranges due to minimal overlap between an image and points from a tilted LiDAR. Our results suggest that our proposed sensor fusion-based SLAM system is a promising solution for challenging outdoor localization and mapping scenarios.

There are several potential avenues for future research related to the PVL-Cartographer SLAM system presented in this study:

- First, by using advanced depth estimation or completion methods, more comprehensive range maps can be created, allowing for the overlay of ranges on additional visual features;
- Second, integrating range measurements into both local and global bundle adjustments could enhance the system's accuracy;
- Third, efforts are underway to upgrade the existing PVL-Cartographer SLAM to a visual-LiDAR-IMU-GPS SLAM system featuring a more tightly integrated pose graph or factor graph;
- Fourth, the system could be expanded by developing a SLAM pipeline that incorporates both visual and LiDAR features;
- Fifth, applying deep neural network techniques for feature classification and pose correction may improve the system's overall performance.
- Finally, exploring the use of the panoramic-vision-LiDAR fusion method in other areas of applications, such as object detection based on RGB imagery acquired by unmanned aerial vehicles (UAVs) [42]. The combination of panoramic images with a broad FOV and the LiDAR data should improve the performance of the transfer-learning-based methods for small-sized object detection.

**Author Contributions:** Conceptualization, Y.Z., J.K. and G.S.; Methodology, Y.Z. and J.K.; Software, Y.Z. and J.K.; Validation, Y.Z. and J.K.; Formal analysis, Y.Z.; Writing—original draft, Y.Z.; Writing—review & editing, J.K. and G.S.; Supervision, G.S.; Project administration, G.S.; Funding acquisition, G.S. All authors have read and agreed to the published version of the manuscript.

**Funding:** This research was funded by "Natural Science and Engineering Research Council of Canada–NSERC (grant no. CRDPJ 537080-18)".

**Data Availability Statement:** The data presented in this study are available upon request to the corresponding author.

**Conflicts of Interest:** The authors declare no conflict of interest.

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
