# Peer review of "PVL-Cartographer: Panoramic Vision-Aided LiDAR Cartographer-Based SLAM for Maverick Mobile Mapping System"

_remotesensing, doi:10.3390/rs15133383_

Round 1

Reviewer 1 Report

For the mobile mapping system (MMS), a crucial component in generating 3D maps for robotics or autonomous driving, the PVL-Cartographer simultaneous localization and mapping system is proposed and the system is able to collect comprehensive environmental data. Specifically, it incorporates multiple sensors to yield dependable and precise mapping and localization and early fusion and intermediate fusion strategies are put forward. The methodology has been validated through solid testing results in outdoor scenarios and has shown good accuracy. Based on these merits, it is a good and interesting piece of research work in this field and can be accepted after answering some of my minor questions:

-Is that possible to give the position error in the x and y directions respectively for Figure 7. If so, readers can understand the statistics of the localization performance.

-Please elaborate more on the caption of Figure 3 and 4.

- Multi-sensor including LiDAR, camera, IMU and GPS has been used. To better use them for sensor fusion, how are the sensors calibration? Calibrating these multi-modal sensors is a non-trivial work and is important for either the early fusion or intermediate fusion. I hope the authors can mention this in the paper briefly by using some literature as supports: estimation on imu yaw misalignment by fusing information of automotive onboard sensors.

- When generating the maps, I saw in this work only the camera, imu and lidar data has been fused. Will the GNSS sensor be helpful for generating the map in a large scale as the drift error will inevitably appear when only the odometry based method is used? In: an automated driving systems data acquisition and analytics platform, the Kalman filter was used to fuse the data from lidar and gps when generating the maps. Please include these kind of work in the paper to mention the merits or limitations of the work in the paper.

- It looks like the results are pretty promising and the accuracy is good. In that regard, I think the camera lidar fusion method can also be used tin other areas of application such as the unmanned aerial vehicles: yolov5-tassel: detecting tassels in rgb uav imagery with improved yolov5 based on transfer learning. Please include this work and discuss the advantages and its applicability in other areas.

-When you create the ground truth data in Figure.6. Was the GPS module in RTK status or other differential correction? Please elaborate more on this.

- Please give more information about the groundtruth data. From the paper, the GPS/IMU is just mentioned. How the groundtruth data is collected was not mentioned. Research such as: autonomous vehicle kinematics and dynamics synthesis for sideslip angle estimation based on consensus kalman filter; automated vehicle sideslip angle estimation considering signal measurement characteristic; improved vehicle localization using on-board sensors and vehicle lateral velocity; have contributed to this area. Please discuss these works and mention how the ground truth data is generated.

Author Response

We have attached our response letter to Reviewer 1 in a separate pdf file.

Reviewer 2 Report

This manuscript presented the Panoramic Vision-aided LiDAR Cartographer-based SLAM for Maverick Mobile Mapping

System.

I have several comments as follows,

- The proposed method used an IMU (Inertial Measurement Unit) 10

data. However, in the title that has not included.

- The contribution needs concise and focus on contribution and novelty. What

- The methodology should be presented in more detail and explained more in Fig.3, 4.

- Mathematic equations should be edited with a general format such as scalar, vector, and matrix. The equations should explain their ingredient.

- How can the author synchronize all sensor data?

- The Table 2, 3. The authors should compare with the SOTA method using the same sensor setup or less, such as LIO-SAM, ORM SLAM 3,... with and without loop closure detection.

I have no further comments on this.

Author Response

We have attached our response letter to Reviewer 2 in a separate pdf file.

Reviewer 3 Report

This is a research article that proposes a SLAM based Mobile Mapping System.

Section "2. Related Work" is comprehensive enough to cover the related activities in different scientific branches. Many relevant works have been listed. Two important critics are hold.
a) The related works have been listed in a mixed form. Advantages and disadvantages of the methods against to the proposed method should be discussed. How do you justify the necessity of your proposed work against to the already available methods?
b) There are also many industrial MMS products which are in-use in daily projects. The commercial MMS systems related to the proposed work should be addressed by adding another sub-title. Most of the currently listed literature are the academic prototypes.

The sentence at Line 295 and196 needs a revision.

 Only minor editing of English language is required.

Author Response

We have attached our response letter to Reviewer 3 in a separate pdf file.

Round 2

Reviewer 2 Report

The authors addressed my concerns,

Thank you for your efforts, and I have no further comments

N/A